# CHATEDIT: Towards Multi-turn Interactive Facial Image Editing via Dialogue

**Xing Cui**[1*]   **Zekun Li**[2*]   **Peipei Li**[1†]   **Yibo Hu**[3]

**Hailin Shi**[3]   **Chunshui Cao**[4]   **Zhaofeng He**[1]

[1]Beijing University of Posts and Telecommunications   [2]University of California, Santa Barbara

[3]NIO   [4]WATRIX.AI

{cuixing, lipeipei, zhaofenghe}@bupt.edu.cn   zekunli@cs.ucsb.edu

{huyibo871079699, hailinshi.work}@gmail.com   chunshui.cao@watrix.ai

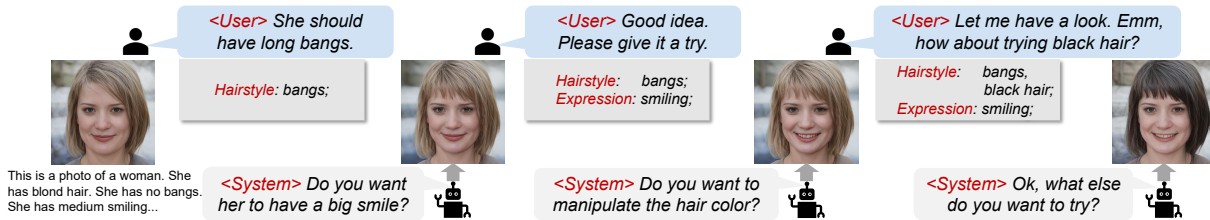

Figure 1: **Illustration of the multi-turn interactive facial image editing task.** The system is required to track the user requests on the facial attributes (in the gray box), edit the image, and generate the response.

## Abstract

This paper explores interactive facial image editing via dialogue and introduces the CHATE-DIT benchmark dataset for evaluating image editing and conversation abilities in this context. CHATEDIT is constructed from the CelebA-HQ dataset, incorporating annotated multi-turn dialogues corresponding to user edit requests on the images. The dataset is challenging, as it requires the system to dynamically track user requests, edit images, and generate appropriate responses. Accordingly, we propose three benchmark tasks: (*i*) user edit request tracking, (*ii*) image editing, and (*iii*) response generation. We present a novel baseline framework that integrates a dialogue module for both tracking user requests and generating responses and an image editing module for image editing. Unlike previous approaches, our framework directly tracks user edit requests from the entire dialogue history up to the current turn and modifies the original image rather than adjusting the previous turn's output, thereby reducing error accumulation and preventing attribute forgetfulness. Extensive experiments on the CHATE-DIT dataset underline our framework's superior performance against prior models, while also highlighting potential room for further research. We will release the code and data publicly to facilitate advancements in complex interactive facial image editing [‡].

---

[*] Equal contribution.

[†] Corresponding Author.

[‡] Our data and codes are available at https://github.com/cuixing100876/ChatEdit

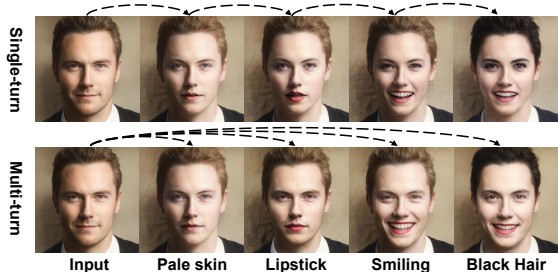

Figure 2: **Comparison of previous repeated single-turn editing approaches and our proposed multi-turn editing approach.** The cascaded errors in the single-turn approach lead to unintended changes in gender and eye makeup.

## 1 Introduction

With the rise of deep generative models such as GANs (Goodfellow et al., 2020; Karras et al., 2019; Li et al., 2023a) and DDPMs (Ho et al., 2020; Sohl-Dickstein et al., 2015; Zhang et al., 2023), significant progress has been achieved in instruction-based facial image editing (Xu et al., 2022; Patashnik et al., 2021; Li et al., 2023b). However, an emerging scenario is multi-turn interactive editing, allowing users to iteratively refine their editing instructions through system interaction (Sharma et al., 2018; Cheng et al., 2020; Kim et al., 2019; Jiang et al., 2021). Existing approaches (Zhou et al., 2022; El-Nouby et al., 2019; Kim et al., 2019) typically treat multi-turn editing as a sequence of successive single-turn edits, leading to issues such as attribute forgetting and error accumulation, as depicted in Fig. 2 (first line). Moreover, these tech-

Table 1: **Comparison of our CHATEDIT with existing works on image editing.**

| Method | Scene | Multi-turn Interaction | System Feedback | Dataset | Text Data | Attributes |
|---|---|---|---|---|---|---|
| TransEditor (Xu et al., 2022) | Facial image | ✗ | ✗ | CelebA-HQ | ✗ | 4 |
| HairCLIP (Wei et al., 2022) | Facial image | ✗ | ✗ | CelebA-HQ | ✗ | 2† |
| StyleCLIP (Patashnik et al., 2021) | Facial image | ✗ | ✗ | CelebA-HQ | ✗ | 8 |
| ChatPainter (Sharma et al., 2018) | Realistic image | ✗ | ✗ | MS COCO | dialogue | - |
| TiGAN (Zhou et al., 2022) | Facial image | ✓ | ✗ | CelebA-HQ | ✗ | 8 |
| Talk-to-Edit (Jiang et al., 2021) | Facial image | ✓ | ✓ | CelebA-dialog | user utterance | 5 |
| CHATEDIT (Ours) | Facial image | ✓ | ✓ | CHATEDIT | dialogue | 21 |

†: HairCLIP only considers hairstyle and hair color. 44 text descriptions are collected for hairstyle and 12 text descriptions for hair color.

niques don't fully harness interactivity and user experience. For example, (Sharma et al., 2018; Cheng et al., 2020; Kim et al., 2019) solely process user inputs without offering natural language feedback, while others (Jiang et al., 2021) rely on rigid, hand-crafted response templates, limiting flexibility and naturalness.

To facilitate the research on multi-turn interactive facial image editing, we introduce a novel benchmark dataset named CHATEDIT. Sourced from the CelebA-HQ dataset (Karras et al., 2018), CHATEDIT enhances a selected set of 12k images with annotated multi-turn dialogues that align with user edit requests for facial images. The annotations include user utterances, system responses, and the user's "belief state", which represents the user edit requests from the beginning of the dialogue to the current turn. Fig. 1 illustrates that success on CHATEDIT necessitates the system to track the user edit requests, edit images based on tracked requests, and provide natural language responses to engage with users. To evaluate the performance of multi-turn interactive editing, we define three tasks: (*i*) user edit request tracking, (*ii*) image editing, and (*iii*) response generation. Correspondingly, we introduce a comprehensive set of metrics that evaluate both response and editing quality.

Based on our benchmark dataset, we propose a baseline framework for multi-turn interactive image editing. Our framework seamlessly integrates a language dialogue module and a visual image editing module. Specifically, we employ an end-to-end task-oriented dialogue (TOD) model to extract the user's image edit requests and generate appropriate responses based on the current conversation context. These tracked user requests are transformed into a text prompt to guide the text-based image editing model, StyleCLIP (Patashnik et al., 2021), in performing the desired manipulations on the input raw image. As illustrated in Fig. 2's second line, our benchmark dataset and the proposed framework enable direct editing on the input raw image instead of cascaded modifications to images from previous

turns, thereby reducing error accumulation and attribute forgetting issues. We perform extensive experiments with diverse settings to investigate the effectiveness of our proposed framework on the CHATEDIT dataset. The results suggest that our proposed framework is superior to the previously prevalent cascaded single-turn editing methods regarding both image editing quality and response diversity.

To sum up, our contributions are three-fold: **(1)** We introduce the CHATEDIT benchmark dataset, which could serve as a valuable resource for advancing research in multi-turn interactive facial image editing. **(2)** We propose a novel framework that seamlessly combines a task-oriented dialogue module and an image editing module. This framework effectively tracks user requests, performs image editing, and generates system responses. Importantly, it addresses the issues of attribute forgetting and error accumulation prevalent in previous methods. **(3)** Through qualitative and quantitative evaluations on the CHATEDIT dataset, we demonstrate the superiority of our proposed multi-turn editing framework over previous cascaded single-turn editing methods. We believe these results not only highlight the strengths of our proposed approach but also spur further exploration in this field.

## 2 Related Work

**Facial Image Editing** Traditional image editing techniques have focused on modifying specific given attributes such as age (Li et al., 2020a, 2019), hairstyle (Wei et al., 2022), or other predefined attributes (Zhou et al., 2022; Li et al., 2020b; Zhang et al., 2022). In recent years, with the development of pre-trained vision-language models (Radford et al., 2021; Li et al., 2022a; Wang et al., 2023), there has been a growing interest in human-computer interaction scenarios (Kottur et al., 2022; Jiang et al., 2021). This leads researchers to explore interactive image editing, where users can dynamically adjust their editing requests through interaction with the system (Kim et al., 2019; Lachmy

Table 2: **Illustration of editable attributes in CHATE-DIT,** which are categorized into four groups (slots).

| Slot | Attribute |
|---|---|
| Expression | *smiling, no smiling, angry, sad* |
| Hair color | *brown hair, blond hair, black hair, gray hair* |
| Hair | *receding hairline, sideburns, bangs, no bangs, mustache, goatee, no beard* |
| Makeup | *no makeup, heavy makeup, lipstick, bushy eyebrows, rosy cheeks, pale skin* |

Table 3: **Statistics of dialogues in CHATEDIT dataset.**

| | |
|---|---|
| Total # dialogues | 12,000 |
| Total # utterances | 96,174 |
| Avg # turns per dialogue | 4.0 |
| Avg # utterances per dialogue | 8.0 |
| Avg # words per user turns | 11.6 |
| Avg # words per system turns | 11.3 |
| Avg # attributes mentioned per dialogue | 6.3 |

et al., 2022; Zhou et al., 2022; Jiang et al., 2021). For instance, TiGAN (Zhou et al., 2022) generates images iteratively based on successive editing steps during a conversation.

However, these methods merely "listen" to user requests without generating system responses, constraining their interaction capability. A recent work called Talk-to-Edit (Jiang et al., 2021) attempts to address this by introducing a rule-based method to generate system response. However, this approach lacks flexibility and struggles with unforeseen scenarios that are not predefined. Furthermore, these methods treat multi-turn editing as a sequence of individual single-turn edits, resulting in issues such as error accumulation and attribute forgetting as the number of interactions increases.

**Task-oriented Dialogue**   There are two primary approaches to TOD. Traditional systems adopt a pipelined approach comprising four modules. Firstly, the natural language understanding (NLU) module (Abro et al., 2022) converts user requests into semantic slots, domain information, and user intention. Secondly, the dialogue state tracking (DST) module (Wu et al., 2019; Le et al., 2019; Lin et al., 2021; Heck et al., 2023) extracts the dialogue state, which records user requests in the form of slot-value pairs. The dialogue policy learning (POL) module (Chen et al., 2017; Geishauser et al., 2022) determines the next action of the dialogue agent based on the dialogue state. Finally, the natural language generation (NLG) module (Elder et al., 2020) generates the system response. In more recent times, there has been a shift towards end-to-end task-oriented dialogue systems (Hosseini-Asl et al., 2020; Wang et al., 2022; Le et al., 2020; Zeng et al., 2023). For example, PPTOD (Su et al., 2022) employs multi-task training that simultaneously processes all sub-tasks. Currently, there have been approaches that utilize Large Language Models (LLMs) like ChatGPT for task-oriented dialogues (Li et al., 2023c). However, existing TOD systems primarily focus on scenarios such as book-ing and consulting, neglecting the interactive image editing scenario.

## 3   CHATEDIT Benchmark Dataset

CHATEDIT is designed to simulate a scenario where a user interacts with a system to manipulate a facial image. Each sample in the CHATEDIT dataset comprises a facial image and a dialogue between the user and the system regarding the editing of the image. Notably, each turn of the dialogue is annotated with the user belief state, representing the user requests that guide both response generation and image editing. In this section, we will first introduce how to construct the dataset and its statistics. Then, we introduce benchmark tasks of CHATEDIT and corresponding evaluation metrics.

### 3.1   CHATEDIT Dataset Construction

**Facial Image Data**   We construct the CHATEDIT dataset on top of the CelebA-HQ (Karras et al., 2018), which is a high-resolution facial image dataset with 30k images. It provides binary annotations of 40 facial attributes. We select 17 attributes from CelebA-HQ and add another four frequently-used attributes "*sad*", "*angry*", "*no smiling*", and "*no bangs*", resulting in a total of 21 editable attributes in the CHATEDIT dataset. As shown in Table 2, these editable attributes are categorized into four groups. We select 12k images from the CelebA-HQ and utilize the annotation in CelebA-HQ to build a caption for the image. An example is presented in Fig. 8 in Appendix.

**Dialogue Annotation**   During each turn of the dialogue, the user expresses their editing requests using natural language. It is essential for the system to detect these requests as belief states and map them to appropriate responses. Consequently, we annotate three types of data for each turn in the dialogue: (1) the user utterance, (2) the user belief state (user requests), and (3) the system response.

We here adopt terminology commonly used in task-oriented dialogue research to introduce the annotations. Each group in Table 2 is considered a

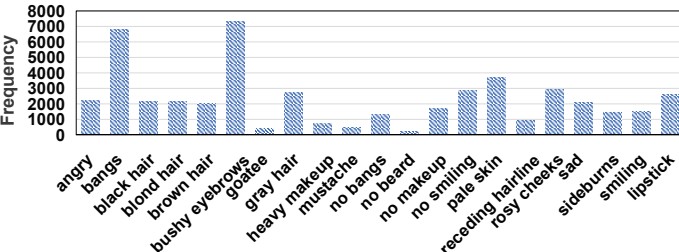
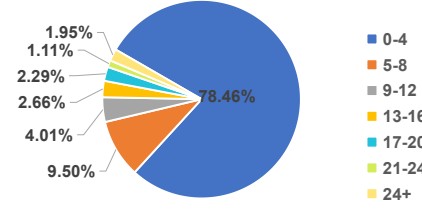

Figure 3: **Analysis of CHATEDIT dataset.** Left: Distribution of different attributes. Right: Distribution of occurrence frequency of various attribute combinations.

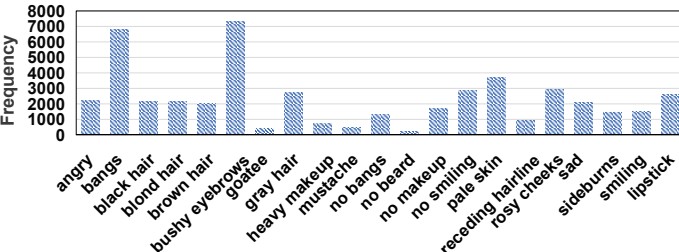

Figure 4: **Illustration of the dialogue flow for the first 3 turns in the CHATEDIT dataset.** User-$n$ and System-$n$ represent the $n$-th user and system turn, respectively.

slot, and each editable attribute represents a possible slot value for its corresponding slot. As a result, the user request in each turn of the dialogue is represented as slot-value pairs. For example, a user request could be "*expression: smiling, hairstyle: bangs, hair color: black hair.*"

To reduce human efforts in annotating the dialogues, we leverage a two-phase pipeline (Karras et al., 2018) consisting of a simulation phase (Li et al., 2022b) and a paraphrase phase. During the simulation phase, we begin by preparing a collection of varied human-written utterances that describe user requests for each editable attribute and system responses for each system action. The supported system actions include *Next*: general queries on what to edit in the next turn, *Request*: request whether to edit an attribute, *Suggest*: suggestions to edit towards a specific attribute value. We then utilize ChatGPT (Ouyang et al., 2022) to generate more diverse utterances by combining task instruction and a few human-written example utterances as the prompt as shown below (Li et al., 2021).

> Write diverse sentences to express the given image editing requirements.
> **Requirements**: smiling
> **Sentence 1**: Can you edit the image to show him with a smile?
> **Sentence 2**: It would be great if you could add a smile to his face.
> ...
> **Sentence N**: I'd love to see a version of the image where he has a big smile on his face.
> **Sentence N+1**:

Next, we construct the dialogue by following a series of steps. Firstly, we determine the user request of the current turn by randomly selecting

editable attributes (excluding the original attributes of the raw image) and expressing them as slot-value pairs. Secondly, we select an utterance that effectively expresses this particular request. Then, based on a predefined policy, we determine the appropriate system action and choose a candidate system response. Specifically, the predefined policy is a set of rules that govern the dialogue flow to ensure it aligns logically with the user's instructions and system functions. For instance, if a user requests a change in hair color to blond, the system's predefined policy would prevent it from generating a redundant or illogical suggestion like "Do you want to dye your hair blond?". Finally, the multi-turn dialogue can be constructed by repeating the above steps.

Following the simulation of multi-turn dialogues, a manual review is conducted to ensure their quality. Human annotators are tasked with reviewing the dialogues to ensure they are following the predefined dialogue flow logic. Additionally, they are asked to refine the expressions to enhance diversity and naturalness if necessary. Fortunately, this process of checking and revising the dialogues is less labor-intensive compared to annotating the dialogues from scratch, resulting in significantly reduced annotation efforts.

### 3.2 CHATEDIT Dataset Statistics

The constructed CHATEDIT dataset comprises 12k examples, with each example consisting of a facial image equipped with a corresponding caption and an annotated multi-turn dialogue. We divide the CHATEDIT dataset into training, validation, and testing sets with 10k, 1k, and 1k examples, respectively. To offer deeper insights into the dataset, we present comprehensive statistics in this section.

**Analyzing Dialogues** Our CHATEDIT dataset comprises a total of 12k dialogues, consisting of approximately 96k utterances. The statistical details of the dataset are presented in Table 3. More-

over, we provide a visual representation of the dialogue flow in Fig. 4. Each block in the figure corresponds to a specific turn labeled as Start, User-$n$, or System-$n$, where $n$ represents the corresponding turn index. For user turns, there are four different kinds of blocks, each representing an attribute group slot. Additionally, there are three types of system turns, each representing a system action. The width of each block in the visualization indicates its occurrence frequency, while the connectivity between blocks signifies their co-occurrences in the dialog flow. Notably, the dataset exhibits a balanced distribution of different dialog flows, indicating a high level of diversity.

**Analyzing Editing Attributes**    Regarding the annotated user requests, we provide insights into the frequency of each attribute (left) and the occurrence frequency of different attribute combinations (right) in Fig. 3. The figure illustrates that the majority of attributes are mentioned over 500 times, while more than 78% of the attribute combinations occur less than 5 times. These findings highlight the diversity present in our dataset.

### 3.3    CHATEDIT Evaluation

To evaluate the system's ability to detect user requests, generate the appropriate system response, and edit the image according to user requests, we propose three benchmark tasks.

**User Request Tracking**    Similar to the dialogue belief state tracking task in TOD, we introduce the user request tracking task to evaluate the system's ability in detecting the user requests. The user requests are represented as slot-value pairs, where each slot denotes a category (group) of attributes and the value indicates a specific attribute value. The input for this task is the dialogue history from the beginning to the current turn. The performance is evaluated with **Joint Accuracy**, which is the ratio of dialogue turns whose slots are predicted completely accurately, i.e., all the slot and slot values are predicted correctly.

**Response Generation**    Another objective of the dialogue module is to generate fluent, reasonable, and diverse responses to interact with the users. We use BLEU (Papineni et al., 2002) to evaluate the fluency of generated responses with respect to reference responses. To evaluate the diversity of generated responses, we employ Distinct-1, 2 (Li

et al., 2015), where Distinct-$n$ represents the ratio of distinct n-grams in responses.

**Image Editing**    Editing an image according to the user request is a primary objective of the CHATEDIT system. Given user requests that consist of multiple attributes, a robust system should be capable of simultaneously manipulating these attributes while maintaining high quality. We evaluate the image editing performance from two perspectives: (1) *relevance*: whether the requested attributes are accurately edited; and (2) *quality*: whether the edited image is realistic and natural.

We measure the editing relevance of each attribute via the cosine similarity of the attribute and the edited image. To evaluate the relevance of multiple attributes, we report two metrics: *Average Relevance (AvgRel)* reflects the average editing relevance on all requested attributes, where the relevance is measured with the cosine similarity between the edited image and the requested attribute:

$$AvgRel = avg\left\{Cos_{\mathrm{CLIP}}\left(I, t\right)\right\}_{t \in T}. \quad (1)$$

$Cos_{\mathrm{CLIP}}$ represents the cosine similarity function. $T$ represents the set of all editing attributes.

We also report *Minimum Relevance (MinRel)*, which reflects the worst relevance among the requested attribute:

$$MinRel = min\left\{Cos_{\mathrm{CLIP}}\left(I, t\right)\right\}_{t \in T}. \quad (2)$$

As for the measure of image quality, we utilize FID (Heusel et al., 2017) and LPIPS (Zhang et al., 2018) metrics, which calculate the statistical similarity between the edited images and the originals.

## 4    Method

In this section, we introduce our proposed framework for CHATEDIT. As illustrated in Fig. 5, the system consists of a task-oriented dialogue module and an image editing module. We adopt a pre-trained language model-based TOD model as the unified dialogue module for both User Request Tracking and response generation. Specifically, it takes the dialogue context prepended with different prompts as input and outputs the tracked user requests and corresponding system response. As for the image editing module, a text-based image editing model StyleCLIP (Patashnik et al., 2021) is utilized, which receives the user requests tracked by the dialogue model as input and edits the input image accordingly.

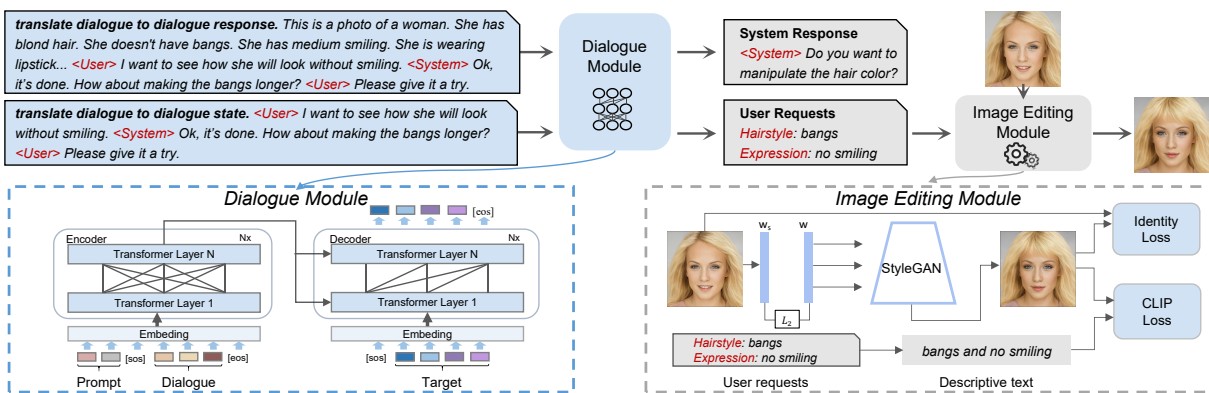

Figure 5: **Illustration of the proposed framework.** The dialogue understanding module is utilized to track user requests and generate system responses. Then, the tracked user requests are fed to the image editing module to guide the image manipulation.

## 4.1 Dialogue Module

The primary goal of the dialogue module is to track user requests and generate natural language feedback as responses. Following (Su et al., 2022), we formulate the two tasks as text generation and adopt a unified pre-trained language model T5 (Raffel et al., 2020) for both two tasks. Specifically, we prepend a task-specific prompt to the dialogue context to serve as the dialogue language model's input, and the model is trained to output the task-specific output. We use "*translate dialogue to dialogue state*" and "*translate dialogue to dialogue response*" as prompts for the User Request Tracking task and the response generation task, respectively. As the facial attribute values of the initial image can provide valuable context to enhance the generation of reasonable conversations, we incorporate the image caption of the initial raw image into the input for the response generation task. Finally, the dialogue model is trained to generate user requests and system responses in a multitask learning approach.

We use the T5 (Raffel et al., 2020) model and initialize it with the PPTOD (Su et al., 2022) checkpoint that has been pre-trained on a diverse set of dialogue corpus and thus equip with primary TOD task completion skills. Given task-specific prompt $z_t$, dialogue history $x$, and target output $y$. The dialogue model is trained with a maximum likelihood objective and the loss function is defined as:

$$\mathcal{L}_\Theta = -\sum_{i=1}^{|y|} log P_\Theta \left( y_i | y_{<i}; z_t, x \right), \quad (3)$$

where $\Theta$ is the model parameters.

## 4.2 Image Editing Module

In this paper, we employ a text-driven image editing method inspired by StyleCLIP (Patashnik et al., 2021). This approach combines the generative capabilities of StyleGAN (Karras et al., 2019) with the joint vision-language representation learned by CLIP (Radford et al., 2021). To generate the manipulated image based on user requests, we utilize a CLIP-based loss, which optimizes the latent code of the input image to align with the directions inferred by the descriptive text of the user requests.

As the tracked user requests are stored in the slot-value pair format, we first construct them into the descriptive text prompt $t$ with templates. Then, we manipulate the image by optimizing the latent code. Specifically, the images are projected to the latent codes and manipulated in the $\mathcal{W}+$ space, which is extended from the $\mathcal{W}$ space proposed in StyleGAN (Karras et al., 2019). For a StyleGAN with 18 layers, $\mathcal{W}+$ space is defined by the cascade of 18 different vectors $[w_1, ..., w_{18}], w_i \in \mathcal{W}$.

As shown in Fig. 5, the input image is first inverted by a fixed StyleGAN inversion model e4e (Tov et al., 2021) to obtain the latent code of the input image $w_s \in \mathcal{W}+$. Then, the latent code $w \in \mathcal{W}+$ is initialized as $w_s$. $w$ is learnable and will be optimized towards the latent code of the edited image. Denote the optimization result of $w$ as $w^*$, $w^*$ can be viewed as the approximation of the latent code of the edited image. Then, taking $w^*$ as input, we can generate the desired edited image via pretrained StyleGAN. In order to optimize $w$, $w$ is first fed to the pretrained StyleGAN to obtain its corresponding image. Thereby, the supervisory information will be delivered via the image. Specifically, to force $w$ to be consistent with

Table 4: **Quantitative results of the dialogue module on user request tracking and response generation.**

| Model | User Request Tracking | Response Generation | | |
|---|---|---|---|---|
| | Joint Acc↑ | BLEU↑ | Distinct-1↑ | Distinct-2↑ |
| T5$_{small}$ | 88.502±0.212 | 14.806±0.203 | 0.393±0.025 | 0.867±0.008 |
| T5$_{base}$ | **89.065**±0.194 | 14.917±0.499 | 0.408±0.005 | 0.872±0.002 |
| T5$_{large}$ | 88.953±0.159 | **16.064**±0.594 | **0.413**±0.010 | **0.886**±0.004 |

the user's descriptive text, the CLIP loss (Radford et al., 2021) is utilized:

$$\mathcal{L}_{\mathrm{CLIP}} = D_{\mathrm{CLIP}}\left(G\left(w\right), t\right), \qquad (4)$$

where $G$ is a pretrained StyleGAN generator that maps latent code into an image. $D_{\mathrm{CLIP}}$ measures the cosine distance between the CLIP embedding of the image and the text. Then, $L_2$ loss is utilized for preserving the similarity between the input image and the edited image in the $\mathcal{W}+$ space:

$$L_2 = \|w - w_s\|_2. \qquad (5)$$

Moreover, $\mathcal{L}_{\mathrm{ID}}$ is used to preserve the identity:

$$\mathcal{L}_{\mathrm{ID}} = 1 - \langle R\left(G\left(w_s\right)\right), R(G(w)) \rangle, \qquad (6)$$

where $R$ is a pretrained face recognition network (Deng et al., 2019), $\langle \cdot, \cdot \rangle$ calculates the cosine similarity. The overall optimization objective is:

$$w^* = \arg \min_{w \in \mathcal{W}+} \mathcal{L}_{\mathrm{CLIP}} + \lambda_{\mathrm{L2}} L_2 + \lambda_{\mathrm{ID}} \mathcal{L}_{\mathrm{ID}}. \qquad (7)$$

# 5 Experiment

## 5.1 Quantitative Evaluation

**Dialogue Module** We initialize our dialogue model with different sizes of pretrained PP-TOD checkpoints PPTOD$_{small}$, PPTOD$_{base}$ and PPTOD$_{large}$, respectively. We further fine-tune the model leveraging the Adam optimizer with a learning rate of 5e-5 and a batch size of 64. We utilize a multi-task training strategy in which the User Request Tracking task and the Response Generation task are trained simultaneously.

As shown in Table 4, we evaluate the dialogue module on the user request tracking and response generation task. All three models achieve a joint accuracy of over 88%, indicating their effectiveness. **We also compare our model with Chat-GPT**. Specifically, we used the prompt proposed by (Heck et al., 2023) to leverage ChatGPT for User Request Tacking, which is up-to-date and has achieved impressive results on the MultiWOZ dataset. Our method achieves a Joint Accuracy of 88.86%, outperforming ChatGPT's 76.26%

Table 5: **Quantitative results of the image editing performance.** Input represents the input of the image editing module. USR, Dial, and USR-T stand for oracle user requests, oracle dialogue, and the user requests tracked by the dialogue module respectively. All experiments utilize StyleCLIP to manipulate images.

| Editing Mode | Input | Image Editing | | | |
|---|---|---|---|---|---|
| | | FID ↓ | LPIPS ↓ | MinRel ↑ | AvgRel ↑ |
| Single-turn | USR | 41.451±0.176 | 0.482±0.002 | 0.730±0.010 | 0.752±0.010 |
| Multi-turn | USR | **40.115±0.165** | **0.449±0.001** | **0.754±0.009** | **0.773±0.009** |
| Multi-turn | Dial | 42.813±0.161 | 0.477±0.001 | 0.741±0.007 | 0.761±0.006 |
| Multi-turn (Ours) | USR-T | 40.536±0.105 | 0.449±0.001 | 0.753±0.010 | 0.773±0.010 |

As for the performance regarding the response generation, an improvement can be seen in both the BLEU score and the Distinct-1,2 score are observed as the model size increases, demonstrating that the larger model can generate responses with better quality and diversity.

**Image Editing Module** The weight of $\lambda_{\mathrm{L2}}$ and $\lambda_{\mathrm{ID}}$ are 0.008 and 0.005, respectively. The editing step is 300, and the Adam optimizer is used with a learning rate of 0.1. For fairness, all experiments utilize StyleCLIP to manipulate images.

To evaluate the effectiveness of our proposed multiple-turn editing approach, we first compare it with single-turn image editing. To eliminate the impact of misidentification of user edit requirements, the oracle user requests are utilized as the input of the image editing module. As shown in the first two lines in Table 5, significant improvements can be observed in all four metrics, indicating that our method successfully mitigates the error accumulation problem and achieves higher image quality. Moreover, we find that the MinRel score of the single-turn method is considerably lower than that of the multi-turn method. This is expected since the single-turn editing approach tends to forget some edited attributes from earlier turns, resulting in suboptimal performance in the poorest attribute.

The last line in Table 5 presents the results of image editing using our proposed pipeline, where the image editing module edits the image based on the user requests tracked by our dialogue module. Notably, the performance achieved in this setting is comparable to the results when oracle user requests are taken as input. This finding highlights the effectiveness of our pipeline, particularly in real-world applications where obtaining perfect oracle user requests may not be feasible.

**Comparison with Talk-to-Edit** We conduct a quantitative comparison between our method and

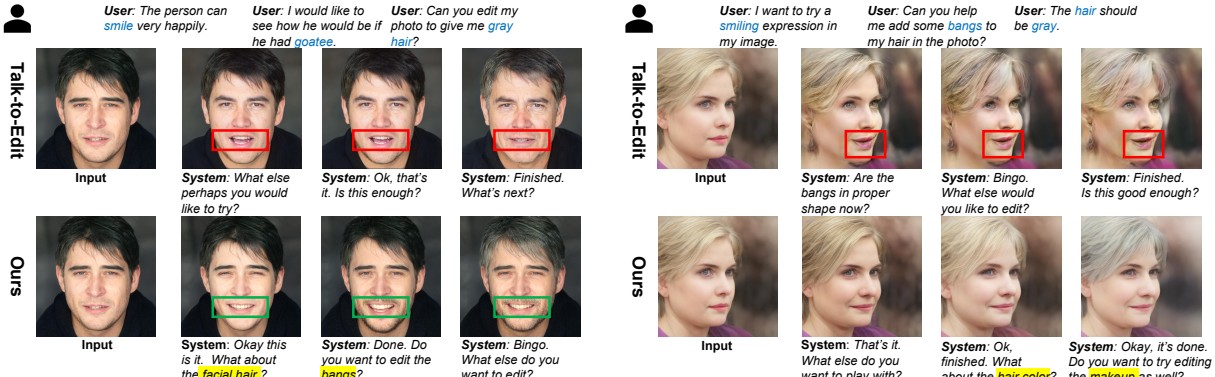

Figure 6: **Visualization of the comparison between our method with Talk-to-Edit.** Left shows the attribute forgetting problem of Talk-to-Edit, where the *smiling* attribute is gradually diluted. Right shows the error accumulation problem. Besides, our method can generate instructive responses by giving suggestions (highlighted in yellow).

Table 6: **Quantitative comparison of the image editing performance between ours and Talk-to-Edit.**

| Model | FID ↓ | LPIPS ↓ | MinRel ↑ | AvgRel ↑ |
|---|---|---|---|---|
| Talk-to-Edit | 132.555 | 0.644 | 0.731 | 0.739 |
| Ours | 98.760 | 0.439 | 0.770 | 0.776 |

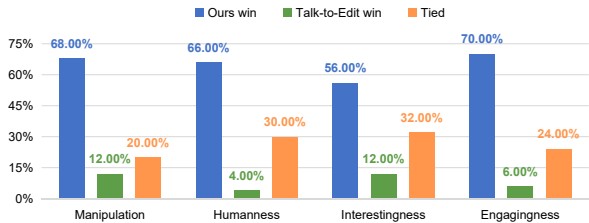

Figure 7: **Human evaluation of ours v.s. Talk-to-Edit** on one axis on image editing: *Manipulation* and three axes on response generation: *Humanness*, *Interestingness*, and *Engagingness*.

the representative single-turn approach Talk-to-Edit on 100 randomly selected samples. As shown in Table 6, our framework achieves better image quality with lower FID and LPIPS scores, highlighting reduced error accumulation. Moreover, our method excels in MinRel and AvgRel metrics, underscoring its better alignment with user requests and effective mitigation of attribute forgetting.

**Tracked user request vs Dialogue** To better understand the essentials of our introduced dialogue module, we experimented with using the raw dialogue context instead of the extracted user requests as input for the image editing module. Table 5 reveals that while the MinRel and AvgRel scores from this ablation study remain competitive with our approach, the FID and LPIPS scores drop notably. This discrepancy can be attributed to the presence of noisy information in the dialogue context. These results underscore the importance of accurately extracting user requests from dialogues through our proposed dialogue module. Further visual illustrations can be found in Fig. 13 in the Appendix.

## 5.2 Qualitative Evaluation

In Fig. 2, we compare our multi-turn approach with the single-turn approach, illustrating the attribute forgetting and error accumulation problems of the single-turn approach. Additional visualization results are shown in Fig. 10 in the Appendix. We also

present examples of manipulation results comparing our method with Talk-to-Edit in Fig. 6, showcasing the high image editing quality and response generation diversity achieved by our method. More visualization results can be found in Fig. 14, 15, and 16 in the Appendix.

## 5.3 Human Evaluation

To more comprehensively assess our method's response generation and image editing capabilities compared with Talk-to-Edit (Jiang et al., 2021), we conduct human evaluations. We choose 20 random images and initiate multi-turn interactive facial image editing dialogues using both our method and Talk-to-Edit. Five English-fluent graduate students participate in this assessment, conducting pairwise comparisons between the two methods. The evaluations focus on one aspect of the manipulated image: *manipulation*, and three aspects of the generated response: *humanness, interestingness,* and *engagingness*. The final answer for each question is determined by majority voting.

Fig. 7 visualizes the evaluation results, underscoring the superiority of our method. Specifically, for image editing, our method effectively generates images of high quality, taking advantage of

the extracted concise user requests to guide the image editing process. For response generation, our method generates more human-like, interesting and engaging responses, improving the interactivity of the system. By contrast, Talk-to-Edit relies on a rule-based approach to generate template responses and employs a cascaded single-turn image editing approach, limiting its performance.

## 6 Conclusion

This paper introduces the CHATEDIT benchmark dataset, which we believe could facilitate the research on multi-turn interactive facial image editing. The dataset poses significant challenges as it requires systems to accurately track user requests from dialogues, perform image editing based on these requests, and generate appropriate responses. We propose a baseline framework that seamlessly combines a task-oriented dialogue module and an image editing module. The introduction of the task-oriented dialogue not only enables interaction with users but also extracts concise user requests from the dialogue context to direct the image editing, avoiding the attribute forgetting and error accumulation issues in previous single-turn methods. The empirical results highlight the efficiency of our approach and the potential for further advancements in this exciting research area.

## Limitations

Our work is the first benchmark dataset to explore multi-turn interactive facial image editing via dialogue and establishes baseline performance for a variety of scenarios. However, there is room for improvement in the following aspects: 1) In the dataset construction, we consider 21 attributes as editable attributes. However, there are out-of-domain attributes users might want to manipulate. In this case, the dialogue understanding may neglect the user requests or generate extra hallucinations. 2) The proposed baseline model has two stages, which leverages the powerful capabilities of existing models via lightweight fine-tuning. However, both the dialogue understanding module and the image editing module limit the quality of the manipulated image. This issue might be alleviated by training the whole model end-to-end, which will be included in our future research. In addition, other issues, such as how to construct a more generalized and robust facial image editing model, also require further exploration.

## Ethics Statement

It is important to clarify that the facial images used in the CHATEDIT dataset are selected from CelebA-HQ (Karras et al., 2018), which is a dataset derived from CelebA (Liu et al., 2015). CelebA consists of images collected from the internet and is publicly available for research purposes only. The images in CelebA-HQ have undergone additional post-processing. The dialogues in the CHATEDIT dataset do not contain sensitive or private information. The dataset has been carefully curated to ensure the privacy and confidentiality of individuals. Furthermore, participants involved in the manual paraphrase and human evaluation processes were compensated with reasonable wages.

## Acknowledgement

This work is supported by Major Technology Innovation Program of Hangzhou, China (Grant 2022AIZD0154), National Natural Science Foundation of China (Grant No. 62306041, No. U21B2045), Beijing Nova Program (Grant No. Z211100002121106, 20230484488), and National Key R&D Program of China (Grant No. 2022YFF1202400).

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

# Appendix

## A  CHATEDIT Benchmark Dataset

Each sample in the CHATEDIT dataset contains an input image to be edited with the corresponding caption, the associated dialogue, and the user requests. Notably, similar to most image editing datasets, the dataset does not include ground truth manipulated images for each turn. Fig. 8 presents an example in CHATEDIT that contains a four-turn interaction. Fig. 9 presents the web page for human paraphrasing, where the annotators are required to check and revise the dialogue to make the conversation fluent, natural, and consistent with the user requests.

## B  Network Architecture

### B.1  Architecture of the Dialogue Module

The model used in the dialogue module is based on the pre-trained language model T5 (Raffel et al., 2020), which is a Transformer (Vaswani et al., 2017) encoder-decoder framework. Each Transformer layer comprises an attention mechanism and a feed-forward network. Specifically, the attention mechanism is self-attention in the encoder layer and encoder-decoder attention in the decoder layer. The feed-forward network consists of a dense layer with an output dimensionality of $d_{\mathrm{ff}}$ followed by a ReLU nonlinearity and another dense layer.

For the base model, both the encoder and decoder consist of 12 layers, where the "key" and "value" matrices of all attention mechanisms have an inner dimensionality of $d_{\mathrm{kv}} = 64$ and all attention mechanisms have 12 heads. The output dimensionality of the first feed-forward network in each block is $d_{\mathrm{ff}} = 3,072$ and the dimensionality of all other sub-layers and embeddings is $d_{\mathrm{model}} = 768$. The small model scales the base model by using 6 layers for the encoder and decoder. For each layer, it utilizes 8-headed attention, $d_{\mathrm{ff}} = 2,048$, and $d_{\mathrm{model}} = 512$. The large model has 24 layers for the encoder and decoder. It scales the base model up by using 16-headed attention, $d_{\mathrm{ff}} = 4,096$, and $d_{\mathrm{model}} = 1,024$. Table 7 summarizes the statistics of three models with different sizes.

### B.2  Architecture of the Image Editing Module

To supplement the description of the image editing module, the architecture of StyleGAN2 (Karras et al., 2020) generator is described in detail in this

Table 7: **Variants of Dialogue Module.**

| Model | #Layers | #Heads | $d_{\mathrm{kv}}$ | $d_{\mathrm{ff}}$ | $d_{\mathrm{model}}$ |
|---|---|---|---|---|---|
| Small | 6 | 8 | 64 | 2048 | 512 |
| Base | 12 | 12 | 64 | 3072 | 768 |
| Large | 24 | 16 | 64 | 4096 | 1024 |

Table 8: The breakdown of the StyleGAN2 (Karras et al., 2020).

| $\mathcal{W}+$ layer index | Resolution | Layer name | # Channels |
|---|---|---|---|
| 0 | 4×4 | Conv | 512 |
| 1 | 4×4 | ToRGB | 512 |
| 2 | 8×8 | Conv0_up | 512 |
| 3 | 8×8 | Conv1 | 512 |
| 3 | 8×8 | ToRGB | 512 |
| 4 | 16×16 | Conv0_up | 512 |
| 5 | 16×16 | Conv1 | 512 |
| 5 | 16×16 | ToRGB | 512 |
| 6 | 32×32 | Conv0_up | 512 |
| 7 | 32×32 | Conv1 | 512 |
| 7 | 32×32 | ToRGB | 512 |
| 8 | 64×64 | Conv0_up | 512 |
| 9 | 64×64 | Conv1 | 512 |
| 9 | 64×64 | ToRGB | 512 |
| 10 | 128×128 | Conv0_up | 512 |
| 11 | 128×128 | Conv1 | 256 |
| 11 | 128×128 | ToRGB | 256 |
| 12 | 256×256 | Conv0_up | 256 |
| 13 | 256×256 | Conv1 | 128 |
| 13 | 256×256 | ToRGB | 128 |
| 14 | 512×512 | Conv0_up | 128 |
| 15 | 512×512 | Conv1 | 64 |
| 15 | 512×512 | ToRGB | 64 |
| 16 | 1024×1024 | Conv0_up | 64 |
| 17 | 1024×1024 | Conv1 | 32 |
| 17 | 1024×1024 | ToRGB | 32 |

section. Specifically, it generates images gradually from low resolution to high resolution. Every major layer (every resolution) of the StyleGAN2 generator consists of two types of convolutional blocks: feature space convolutions (Conv), which are leveraged for feature map synthesis, and toRGB convolutions (ToRGB), which utilize convolutions to convert the feature map into an RGB image. Each of these convolution blocks is modulated by a vector of style parameters $w$. In our experiment, we utilized the $\mathcal{W}+$ space where each $\mathcal{W}+$ layer has its own style parameters $w_i \in \mathcal{W}$. The details of StyleGAN2 (Karras et al., 2020) generator are listed in Table 8.

## C  Details of the Human Evaluation

**Questions of the Human Evaluation.** In the human evaluations, we compare our framework with Talk-to-Edit over one aspect of the manipulated image: manipulation, and three aspects on the generated response of each turn in the dialogues: humanness, interestingness, and engagingness. The instructions for these four aspects provided to participants are shown as follows:

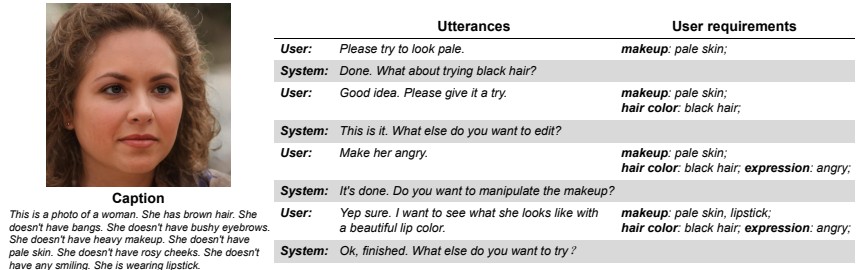

Figure 8: **An example sample from the CHATEDIT dataset.**

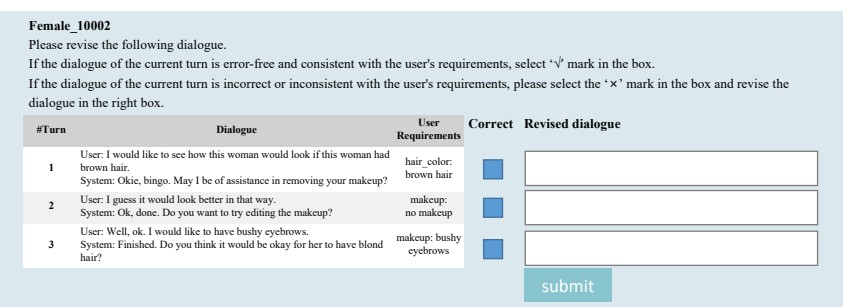

Figure 9: **Interface for the manual paraphrase.**

- Manipulation: *Which one manipulates the image better with facial identity unchanged?*
- Humanness: *Which one sounds more natural and personable?*
- Interestingness: *Which one arouses your curiosity or tells you something new or useful?*
- Engagingness: *Which one is more likely to capture your attention and make you want to further interact with it?*

We conduct the blind evaluation where participants will not be informed about the source of the manipulated images and generated responses (our framework or Talk-to-edit) to ensure fairness.

**Inter-rater agreement.** To evaluate the agreement among the answers of all participants, we calculate the inter-rater agreement score. The average inter-rater agreement score is 0.26 in terms of Fleiss' kappa (Fleiss and Cohen, 1973), which demonstrates a fair agreement.

## D More Qualitative Results

**Comparison of Single-turn Editing and Our Proposed Multi-turn Editing.** In Fig. 10, 11, 12, we present more visualization results to illustrate the attribute forgetting problem and error accumulation problem of previously single-turn methods, which can be avoided by our multi-turn approach. Specifically, we utilize StyleCLIP as the image editing method for both the sing-turn method and our multi-turn method for fair comparison. Notably, as shown in In Fig. 11, 12, our method still performs

better in the setting that the single-turn method takes oracle user requests as input while ours take the dialogue as input and uses the dialogue module to obtain the tracked user requests.

**Visualization Results of Ablation study.** In the ablation study of the main paper, we demonstrate the effectiveness of the introduced dialogue module that extracts user requests to guide the image editing module quantitatively. We present qualitative results in this section. We experiment with the multi-turn method that isn't equipped with a dialogue module to extract the user requests and thus directly takes dialogue as input. As shown in Fig 13, it fails to understand the user request accurately and ignores some of the user requests, suggesting the significance of our introduced dialogue module in the multi-turn interactive editing.

**Comparison Between Our Method with Talk-to-Edit.** More qualitative comparisons between our method with Talk-to-Edit (Jiang et al., 2021) are presented in Fig. 14, 15, 16. Fig. 14 shows that there exists the attribute forgetting problem in Talk-to-Edit. Fig. 15 illustrates that Talk-to-Edit has an error accumulation problem. Besides, as shown in these results, our method can generate better responses with proper suggestions, which improves interactivity. Moreover, error judgment in the rule-based method will lead to the unexpected break-off of the interaction in Talk-to-Edit, which is represented in Fig 16.

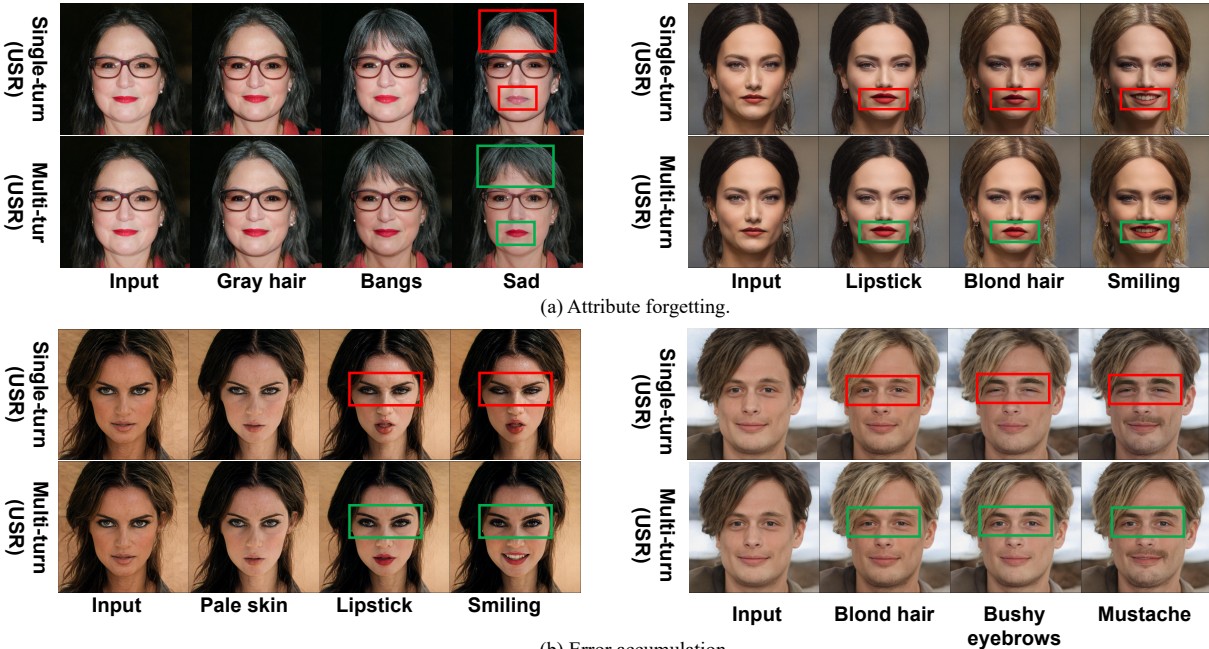

Figure 10: **Comparison of single-turn editing and our proposed multi-turn editing.** Both methods take oracle user requests as input to make a fair comparison. (a) shows the attribute forgetting problem of the single-turn editing method. (b) illustrates the error accumulation problem. By contrast, our proposed multi-turn editing approach avoids these issues.

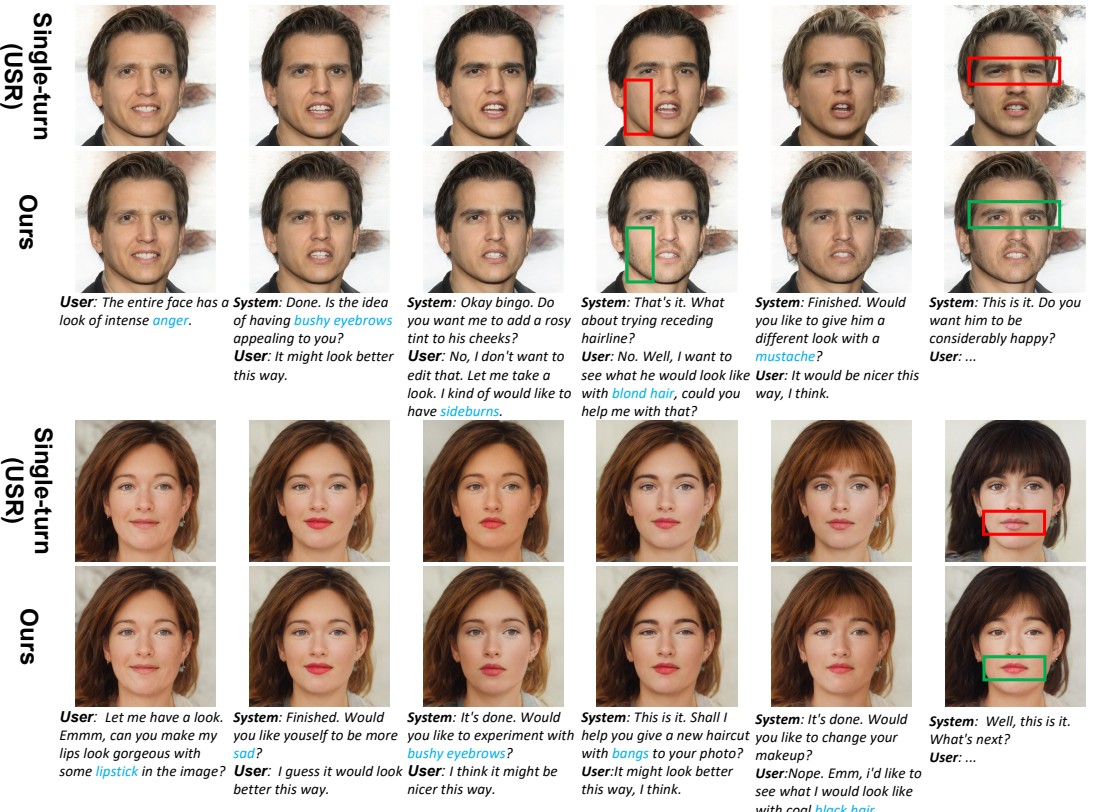

Figure 11: **Comparison of single-turn editing and our proposed framework (a).** Our method takes dialogue history as input and uses the dialogue module to track user requests while the oracle user requests are used for single-turn methods.

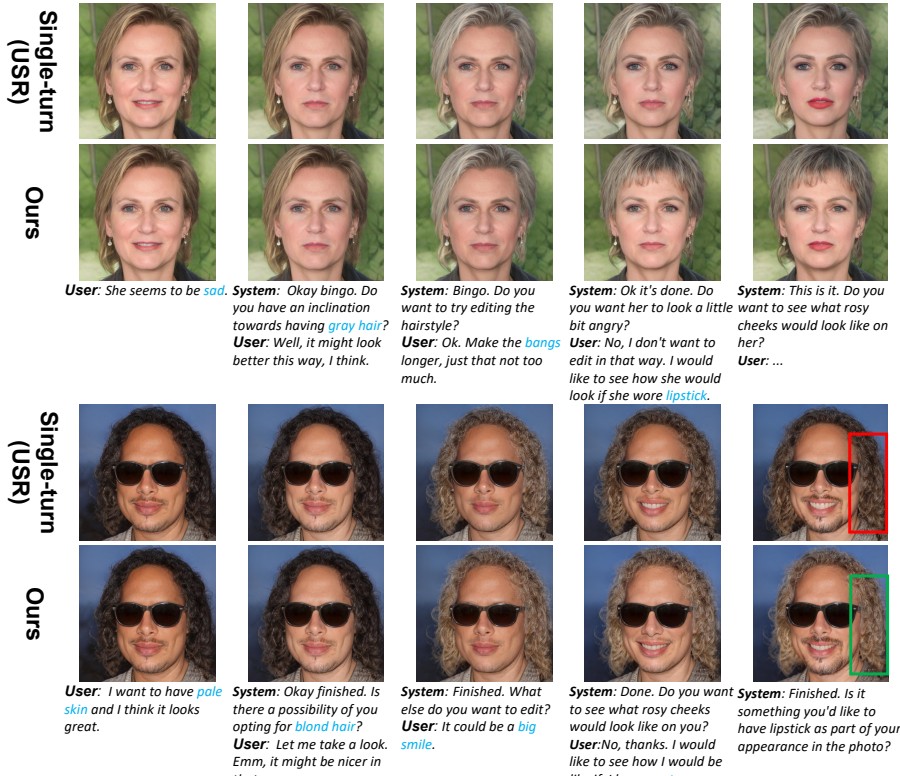

Figure 12: **Comparison of single-turn editing and our proposed framework (b).** Our method takes dialogue history as input and uses the dialogue module to track user requests while the oracle user requests are used for single-turn methods.

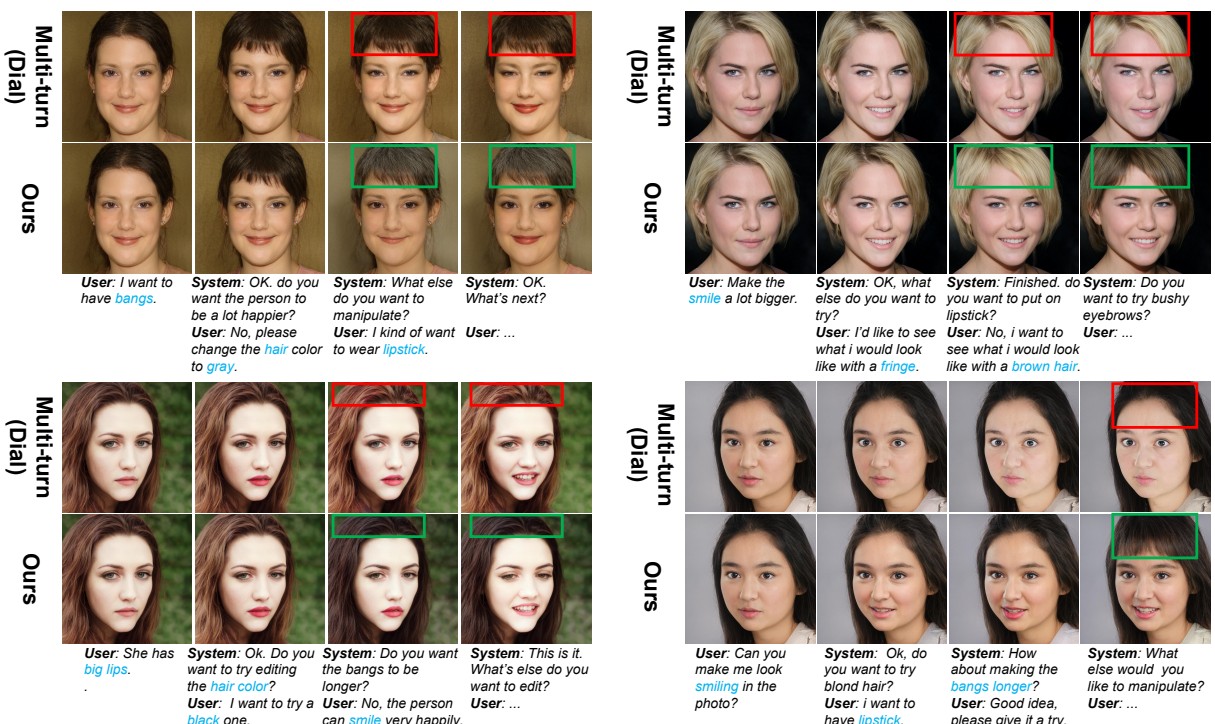

Figure 13: **Visualization results of ablation study.** The cases show the influence of the introduced dialogue module that extracts user requests from the dialogue on the manipulated images. The multi-turn approach doesn't equip with the dialogue module and thus takes the whole dialogue as input of the image editing module.

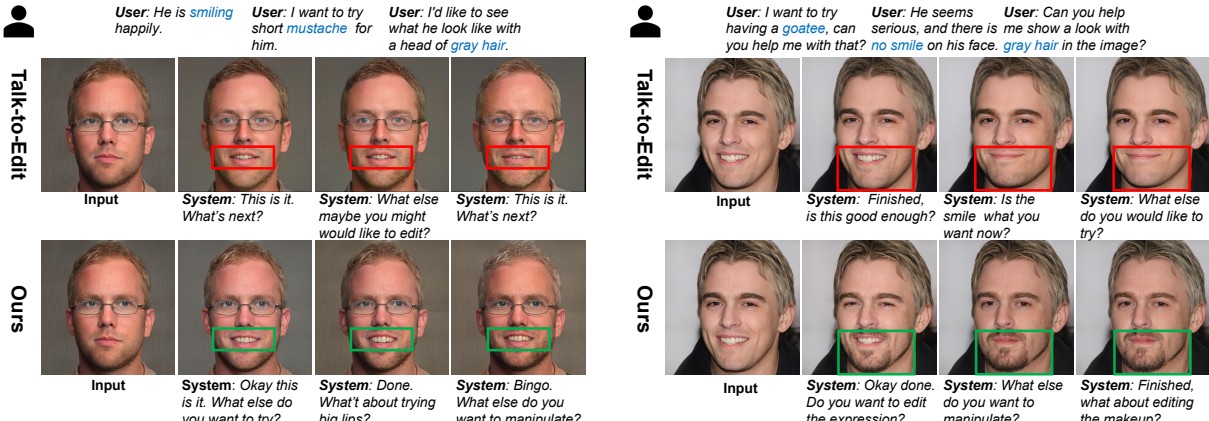

Figure 14: **Comparison of our method and Talk-to-Edit.** These results demonstrate the attribute forgetting problem in Talk-to-Edit, where the goatee attribute is lost in the above cases.

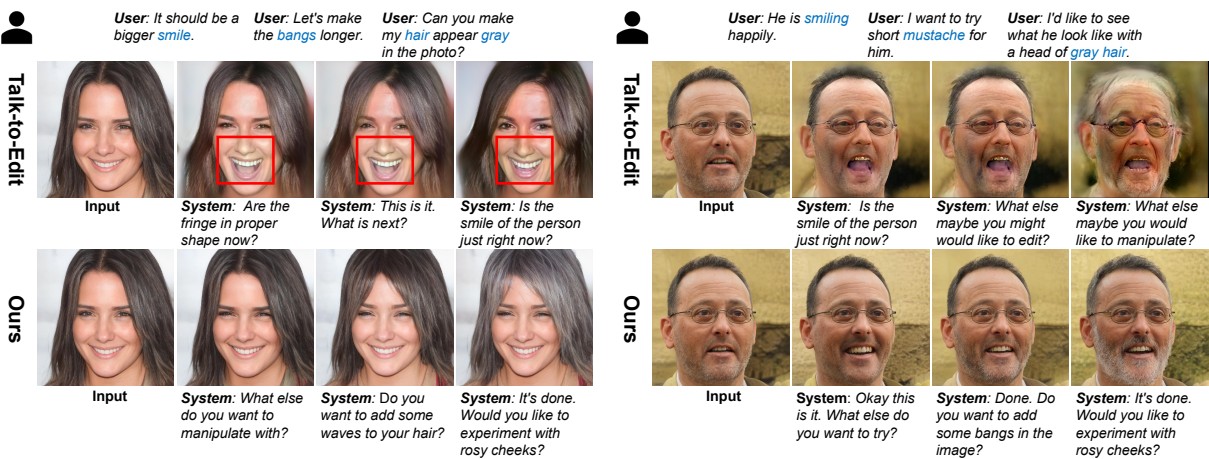

Figure 15: **Comparison of our method and Talk-to-Edit.** These results demonstrate the error accumulation problem in Talk-to-Edit, where artifacts occur and propagate to the final edited image.

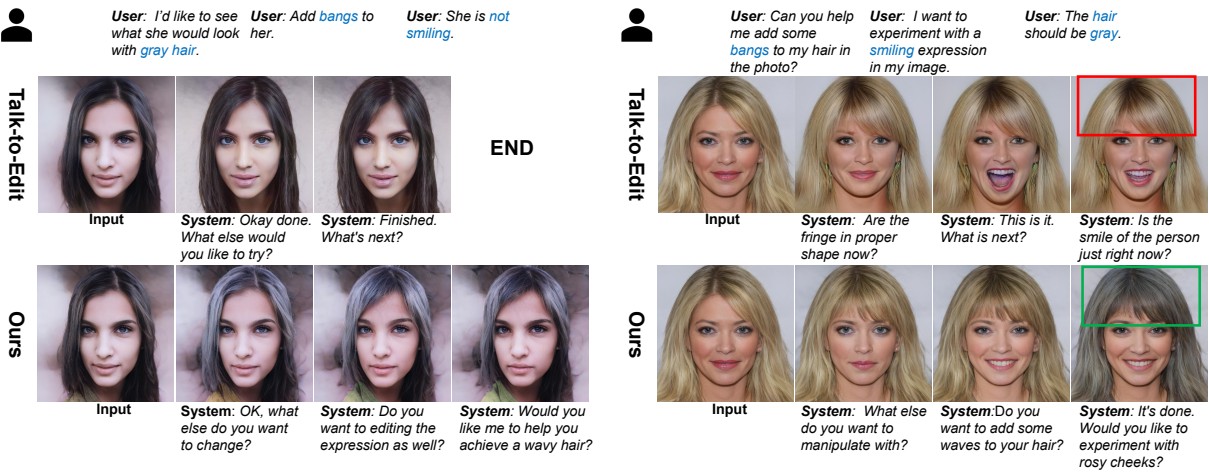

Figure 16: **Comparison of our method and Talk-to-Edit.** The left present case where the interaction is broken off unexpectedly in Talk-to-Edit due to its rule-based method. *END* represents that the system terminates the interaction. The right present case where Talk-to-Edit fails to edit.

Table 9: **Illustration of out-of-distribution attributes.**

| Slot | Attribute |
|---|---|
| Expression | *disgust, surprise, fear* |
| Hair color | *pink hair, purple hair, red hair* |
| Makeup | *big eyes* |

Table 10: **Image editing performance on in-domain and out-of-domain test sets.**

| Test set | FID ↓ | LPIPS ↓ | MinRel ↑ | AvgRel ↑ |
|---|---|---|---|---|
| In-domain | 40.115 | 0.449 | 0.754 | 0.773 |
| Out-of-domain | 40.264 | 0.458 | 0.739 | 0.759 |

# E  More Quantitative Results

## E.1  Out-of-distribution Generalization

A distinct advantage of our proposed framework is its adaptability in utilizing various training strategies for the T5-based dialogue module. Drawing inspiration from (Lin et al., 2021), we cast the slot value tracking as a question-answering task, which promotes enhanced generalization on unseen data. For example, the model is prompted with queries like "what is the hair color?"

For evaluation, we use an out-of-distribution (OOD) test set comprising 1,000 samples based on previously unseen values, as detailed in Table 9. Experimental results suggest our model's robustness against OOD scenarios during testing. The dialogue module achieves an accuracy of 58.90% on the OOD test set. As for image editing, the results on the OOD test set align closely with those of the in-domain test set.

## E.2  Ablation Study of Image Editing Module.

We performed ablation studies for both identity loss ($L_{id}$) and $L_2$ loss within the image editing module. The results are presented in Table 11. It is observed that both loss types influence the image-editing performance, with $L_2$ loss having a more significant impact. This is likely because $L_2$ loss constrains the degree of feature vector transformations in hidden space, and significant changes in StyleGAN's hidden space can introduce severe artifacts.

Table 11: **Ablation study of image editing module.**

| Method | Input | Image Editing | | | |
|---|---|---|---|---|---|
| | | FID ↓ | LPIPS ↓ | MinRel ↑ | AvgRel ↑ |
| Multi-turn | USR | **40.115** | **0.449** | **0.754** | **0.773** |
| w/o $L_{id}$ | USR | 40.475 | 0.452 | 0.742 | 0.763 |
| w/o $L_2$ | USR | 74.639 | 0.794 | 0.715 | 0.752 |