# OpenReview forum: "ChatEdit: Towards Multi-turn Interactive Facial Image Editing via Dialogue"
_EMNLP/2023/Conference — EMNLP 2023 Main_

### Official Review · Reviewer_Ht4U · 2023-08-04

**Soundness:** 3

**Excitement:**

4: Strong: This paper deepens the understanding of some phenomenon or lowers the barriers to an existing research direction.

**Missing References:**

UniConv: A Unified Conversational Neural Architecture for Multi-domain Task-oriented Dialogues, EMNLP 2020

Non-Autoregressive Dialog State Tracking, ICLR 2020

**Paper Topic And Main Contributions:**

The authors propose a new task and benchmark for interactive facial image editing through dialogues. A new benchmark called ChatEdit is created to evaluate AI models in image editing and conversation response generation. The benchmark contains useful annotation to track user requests and generate responses.

A new model is also proposed to perform this task end-to-end, including a dialogue module and an image editing module.


**Questions For The Authors:**

- L244: what is “predefined policy”? I think this detail is important and should be described sufficiently

**Reasons To Accept:**

- Large-scale benchmark of 12k images, using the annotation from the Celeb-HQ dataset. Comprehensive analysis of the dataset, including a manual review to ensure the dialogue flow logic.
- The proposed approach is more robust by tracking the user requests from the entire dialogue history in the form of state tracking. This can help to avoid error cascading and accumulation over multiple turns of interactions

**Reasons To Reject:**

- The editable attributes are still quite limited to only 4 types of slots (expression, hair color, hair, and makeup). There are many variances in the real-world and many slots would be missing in the annotations. I wonder if the current approach is still robust with unseen slots/ slot values during test time.
- Additional study could be provided e.g. to test the model in a zero-shot setting with unseen attributes. More ablation could be done (e.g. by model hyperparameters, or losses) to make the results more convincing.

**Reproducibility:**

4: Could mostly reproduce the results, but there may be some variation because of sample variance or minor variations in their interpretation of the protocol or method.

**Reviewer Confidence:**

4: Quite sure. I tried to check the important points carefully. It's unlikely, though conceivable, that I missed something that should affect my ratings.

**Typos Grammar Style And Presentation Improvements:**

- In Figure 2, It is quite difficult to see the difference in generated images between single-turn and multi-turn. What type of error is cascaded in the single-turn and not in the multi-turn approach?

---

> ### Author Rebuttal · Authors · 2023-08-29
>
> Thanks for your valuable comments.
> > **Q1** *If the current approach is still robust with unseen slots/ slot values during test time.*
>
> An advantage of our proposed approach is its flexibility in adopting different training strategies to train our T5-based dialogue module. Following [r2], we formulate the slot value tracking as question answering task to enable better generalization performance on unseen data. Specifically, we will ask the model for the value for each slot, such as "what is the hair color?''. Our tests reveal relative robustness against out-of-distribution (OOD) scenes during test time.
>
> Specifically, we evaluate the out-of-domain performance across various scenarios as detailed in Table r2. We construct a new OOD test set containing 1000 samples based on these unseen values shown in Table r2. For the dialogue module, without modifying our model, 58.90% accuracy is achieved on the OOD test set. For image editing, as shown in Table r3, the performance on the OOD test set is comparable to that of the in-domain test set.  There is still room for improvement in performance on the OOD test set. As discussed in the Limitations section of the main paper, our future research will explore to make an improvement in this topic.
>
> **Table r2: Illustration of added out-of-distribution attributes.**
> Slot     | Attribute
> -------- | --------
> Expression  | disgust, surprise, fear
> Hair color  | pink hair, purple hair, red hair
> Makeup   | big eyes
>
> **Table r3: Quantitative comparison of the image editing performance between in-domain and out-of-domain test sets.**
> Test set | FID↓ | LPIPS↓ | MinRel↑ | AvgRel↑
> -------- | -------- | -------- | -------- | --------
> In-domain | 40.115 | 0.449 | 0.754 | 0.773
> Out-of-domain | 40.264 | 0.458 | 0.739 | 0.759
>
> Reference
>
> [r2] Lin, Zhaojiang, et al. 2021. Zero-Shot Dialogue State Tracking via Cross-Task Transfer. In EMNLP.
>
> > **Q2** *Additional study could be provided.*
>
> We performed ablation studies for both identity loss (Lid) and L2 loss within the image editing module. The results are presented in Table r4. It is observed that both loss types influence the image-editing performance, with L2 loss having a more significant impact. This is likely because L2 loss constrains the degree of feature vector transformations in hidden space, and significant changes in StyleGAN's hidden space can introduce severe artifacts.
>
>  **Table r4: Ablation study.**
> Method | Input | FID↓ | LPIPS↓ | MinRel↑ | AvgRel↑
> -------- | -------- | -------- | -------- | -------- | --------
> Multi-turn | USR | 40.115 | 0.449 | 0.754 | 0.773 |
> w/o Lid | USR | 40.475 | 0.452 | 0.742 | 0.763
> w/o L2 | USR | 74.639 | 0.794 | 0.715 | 0.752
>
>
> > **Q3** *What is “predefined policy”?*
>
> The term "predefined policy" refers to a set of rules that govern the dialogue flow to ensure it aligns logically with the user's instructions and system functions. For instance, if a user requests a change in hair color to blond, the system's predefined policy would prevent it from generating a redundant or illogical suggestion like "Do you want to dye your hair blond?". This policy serves to prevent unrealistic or redundant conversation scenarios. We appreciate the call for clarity and will refine this section in our paper accordingly.
>
> > **Q4** *Missing References.*
>
> Thanks for your valuable suggestion. We will incorporate the suggested citations into the relevant sections of our paper in the updated version.
>
> > **Q5** *Typos grammar style and presentation improvements.*
>
> Thank you for pointing out the difficulty in interpreting Figure 2. In the single-turn approach, the cascaded errors in the single-turn mainly involve unintended changes to the gender and eye makeup. We will aim to improve the clarity of these illustrations in the updated version of our paper.

---

### Official Review · Reviewer_zAak · 2023-08-05

**Soundness:** 3

**Excitement:**

3: Ambivalent: It has merits (e.g., it reports state-of-the-art results, the idea is nice), but there are key weaknesses (e.g., it describes incremental work), and it can significantly benefit from another round of revision. However, I won't object to accepting it if my co-reviewers champion it.

**Paper Topic And Main Contributions:**

This paper introduces an interactive facial image editing tool. This system is constructed with a task-oriented module and an image editing module, and can modify given image following the user requirements in the dialogues.

**Reasons To Accept:**

The proposed system has involved a list of components for user interactive intention understanding, including an adpot T5 dialogue system and ChatGPT, providing reasonble overall quality.

**Reasons To Reject:**

1, for the interactive user intention understanding, ChaGPT could fully handle this task with well-designed prompt, seems the adopt T5 is unneeded.
2, no novelity is notified in the dialogue/image generation system construction.

**Reproducibility:**

4: Could mostly reproduce the results, but there may be some variation because of sample variance or minor variations in their interpretation of the protocol or method.

**Reviewer Confidence:**

4: Quite sure. I tried to check the important points carefully. It's unlikely, though conceivable, that I missed something that should affect my ratings.

---

> ### Author Rebuttal · Authors · 2023-08-29
>
> Thanks for your valuable comments. Please find our response below:
> > **Q1** for the interactive user intention understanding, ChaGPT could fully handle this task with well-designed prompt, seems the adopt T5 is unneeded.
>
> The reasons of the choice of T5 over ChatGPT for User Request Tracking are two-folds:
> - **Performance**: We've conducted a quantitative comparison between our T5-based dialogue model and ChatGPT on 100 randomly selected dialogues. Our method achieves a Joint Accuracy of 88.86%, outperforming ChatGPT's 76.26%. Specifically, we used the prompt settings proposed by Heck et al. [r1], which is up-to-date and has achieved impressive results on the MultiWOZ dataset.
> - **Practical Considerations**: The performance of ChatGPT often hinges on intricate prompt engineering and comes with significant API costs. Additionally, privacy and policy constraints may limit its applicability. Therefore, our model, designed for local deployment, serves as a more practical alternative.
>
> > **Q2** no novelity is notified in the dialogue/image generation system construction.
>
> We introduce a new benchmark dataset for multi-turn interactive image editing along with a baseline for handling this task.
>
> Our approach innovatively integrates a task-oriented dialogue module with an image editing module. Compared with prior work that reduces multi-turn interactions to successive single-turn interactions, our model supports multi-turn engagement and editing. The dialogue module extracts the user edit requests and generates coherent and fluent responses given the full conversation context as input. The extracted user edit requests are derived from the whole conversation instead of turn-level updates, which effectively mitigates issues of **error accumulation** and **attribute forgetting** common in single-turn approaches.
>
> Our experimental results, both quantitative and qualitative, showcase the advantages of our multi-turn framework. The results suggest that our proposed approach serves as an effective baseline for multi-turn interactive image editing tasks.

---

### Official Review · Reviewer_893r · 2023-08-06

**Soundness:** 3

**Excitement:**

3: Ambivalent: It has merits (e.g., it reports state-of-the-art results, the idea is nice), but there are key weaknesses (e.g., it describes incremental work), and it can significantly benefit from another round of revision. However, I won't object to accepting it if my co-reviewers champion it.

**Paper Topic And Main Contributions:**

The authors present a dataset that is derived from Celeb-QA that supports interactive facial image editing via dialogue. The authors incorporate annotated multi-turn dialog. The authors' dialog state tracking solutions take the overall conversation up to the image editing request turn into account, instead of only looking at a single previous turn. The authors provide ablation studies to show the effectiveness of their approach.

**Reasons To Accept:**

The paper is well written and easy to follow. The interactive multi-turn image editing dataset is a good contribution to the research community. The authors provide limited ablation studies in the main body of the text.

**Reasons To Reject:**

The work seems to be an incremental and marginal improvement on prior work( e.g talk-to-edit). The authors show some qualitative comparison with talk-to-edit in the Appendix, which is somewhat convincing.

**Reproducibility:**

4: Could mostly reproduce the results, but there may be some variation because of sample variance or minor variations in their interpretation of the protocol or method.

**Reviewer Confidence:**

3: Pretty sure, but there's a chance I missed something. Although I have a good feel for this area in general, I did not carefully check the paper's details, e.g., the math, experimental design, or novelty.

---

> ### Author Rebuttal · Authors · 2023-08-29
>
> Thank you for providing your insightful feedback on our paper. We would like to clarify that our work distinguishes from Talk-to-Edit in several significant ways:
>
> 1. *New Benchmark Dataset for Multi-Turn Interactive Image Editing.* The Talk-to-Edit paper focuses on single-turn interactions, largely hindered by the absence of a suitable multi-turn benchmark dataset. We address this limitation by introducing a novel dataset tailored for multi-turn interactive image editing. The dataset includes multi-turn conversation and annotations for image editing. We also provide a set of evaluation criteria for assessing multi-turn interactive image editing performance.
>
> 2. *Enhanced User Engagement and Robustness.* Talk-to-Edit relies on a **rule-based approach to returning templated response for single-turn interactions** and often encounters bugs that lead to abrupt exits. We tested this by randomly selecting 100 dialogues, finding that Talk-to-Edit broke off in 73% of cases (see Fig. 17 in the main paper). In contrast, our framework utilizes a language model-based dialogue module, allowing for more coherent and diverse user interactions. It is also able to more reliably extract user edit requests from the whole conversation instead of from each single turn utterance.
>
> 3. *Mitigation of Error Accumulation and Attribute Forgetting Issues.* Talk-to-Edit treats multi-turn editing as a sequence of successive single-turn edits, which leads to issues like error accumulation and attribute forgetting. Our framework alleviates these issues by closely integrating a dialogue module that extracts current user edit requests from the full multi-turn conversation. These extracted requests then guide the image editing module. We supplement a quantitative comparison between our method and Talk-to-Edit on 100 randomly-selected cases. Experimental results are shown in Table r1. Our framework achieves better image quality with lower FID and LPIPS scores, highlighting reduced error accumulation. Moreover, our method excels in MinRel and AvgRel metrics, underscoring its better alignment with user requests and effective mitigation of attribute forgetting.
>
> **Table r1: Quantitative comparison of the image editing performance between ours and Talk-to-Edit.**
> Model | FID ↓ | LPIPS ↓ | MinRel ↑ | AvgRel ↑
> -------- | -------- | -------- | -------- | --------
> Talk-to-Edit | 132.555 |0.644 |0.731 |0.739
> Ours |98.760 |0.439 |0.770 |0.776

---

### Meta-Review · Area_Chair_L5mF · 2023-09-11

**Recommendation:** 4

**Metareview:**

The work provides a dataset and framework for interactive facial image editing through dialogue.

**Pros**: Reviewers agree the dataset will be a good resource to the community. The majority of reviewers also agree the dialog state tracking solution is also a new (if incremental) contribution that is well defended empirically.

**Cons**: Some reviewers raise concerns that the work is incremental compared to previous research (e.g., talk-to-edit and the dialogue/image generation components).

---

### Decision · Program_Chairs · 2023-10-07

**Decision:**

Accept-Main

**Comment:**

The work provides a dataset and framework for interactive facial image editing through dialogue.

**Pros**: Reviewers agree the dataset will be a good resource to the community. The majority of reviewers also agree the dialog state tracking solution is also a new (if incremental) contribution that is well defended empirically.

**Cons**: Some reviewers raise concerns that the work is incremental compared to previous research (e.g., talk-to-edit and the dialogue/image generation components).